# Using Quantitative Trait Locus Mapping and Genomic Resources to Improve Breeding Precision in Peaches: Current Insights and Future Prospects

**DOI:** 10.3390/plants14020175

**Published:** 2025-01-10

**Authors:** Umar Hayat, Cao Ke, Lirong Wang, Gengrui Zhu, Weichao Fang, Xinwei Wang, Changwen Chen, Yong Li, Jinlong Wu

**Affiliations:** 1The Key Laboratory of the Gene Resources Evaluation and Utilization of Horticultural Crop [Fruit Tree], Ministry of Agriculture, Zhengzhou Fruit Research Institute, Chinese Academy of Agricultural Sciences, Zhengzhou 450009, China; 2Zhongyuan Research Center, Chinese Academy of Agricultural Sciences, Xinxiang 453003, China

**Keywords:** peach, QTL, marker-assisted breeding, genomic selection, CRISPR/Cas9, fruit shape

## Abstract

Modern breeding technologies and the development of quantitative trait locus (QTL) mapping have brought about a new era in peach breeding. This study examines the complex genetic structure that underlies the morphology of peach fruits, paying special attention to the interaction between genome editing, genomic selection, and marker-assisted selection. Breeders now have access to precise tools that enhance crop resilience, productivity, and quality, facilitated by QTL mapping, which has significantly advanced our understanding of the genetic determinants underlying essential traits such as fruit shape, size, and firmness. New technologies like CRISPR/Cas9 and genomic selection enable the development of cultivars that can withstand climate change and satisfy consumer demands with unprecedented precision in trait modification. Genotype–environment interactions remain a critical challenge for modern breeding efforts, which can be addressed through high-throughput phenotyping and multi-environment trials. This work shows how combining genome-wide association studies and machine learning can improve the synthesis of multi-omics data and result in faster breeding cycles while preserving genetic diversity. This study outlines a roadmap that prioritizes the development of superior cultivars utilizing cutting-edge methods and technologies in order to address evolving agricultural and environmental challenges.

## 1. Introduction

As a traditional fruit, peaches (*Prunus persica* L. Batsch) are adored by people worldwide for their delicious flavor as well as their pleasing texture and appearance [1]. In order to develop superior cultivars that meet consumer demands and stay competitive in shifting markets, breeders and growers must understand the genetic foundations of peach fruit morphology [2]. This comprehensive review combines comparative analyses with a close look at the historical evolution and current advancements in QTL mapping for peach fruit morphology in order to shed light on the intricate genetic regulatory mechanisms governing fruit-specific traits in peaches and related crops.

A paradigm shift has occurred in peach breeding as a result of QTL mapping’s emergence as an essential tool for comprehending the complex genetic structure governing fruit shape [3,4]. In this work, we examine the intricate realm of enhanced breeding methods focusing on genomic and marker-assisted selection as breeding process accelerators [5]. Peaches, one of the most popular stone fruits, are primarily grown in the US, China, Italy, and Spain [6], yet maintaining fruit productivity and quality is challenging due to climate variability [7]. It is necessary to develop new cultivars with improved resilience and ideal fruit morphology in order to guarantee consistent production. As consumer preferences shift, the increasing demand for fruits with superior quality, consistent shape, and uniform appearance emphasizes the need for sophisticated breeding techniques [8].

According to D’Agostino and Tripodi [9], genomic selection simplifies the process of identifying the best candidates for breeding by using high-throughput sequencing technologies and advanced statistical models to assess an individual’s genetic composition.

Molecular markers that are closely associated with particular QTLs are used in marker-assisted selection to speed up the selection of peaches with desired traits. Wu et al. [10] assert that research on rice shows how marker-assisted methods can be applied more broadly to a variety of crops, including higher accuracy and efficiency. Its application to peach breeding demonstrates how such approaches can be translated to produce better selection results. The versatility of genome-editing tools and transgenic techniques, especially CRISPR/Cas9, for altering genes linked to important traits has also been demonstrated by the extensive exploration of these technologies in crops such as apple trees [11]. Despite their focus on apples, Wang and Chen [11] highlighted the wider potential of CRISPR/Cas9 in fruit crops, pointing to its potential use in peach breeding to modify the genes controlling fruit morphology. Using this innovative method, peach cultivars with improved fruit quality and commercial viability could be created. As shown in Figure 1, precision peach breeding entails a number of complementary processes including genomic selection (GS), QTL mapping, and CRISPR gene editing. In order to successfully develop new peach cultivars that meet consumer demands and enhance environmental resilience, each step is essential.

### 1.1. Quantitative Trait Locus Mapping (QTL): Revealing Genetic Variation and Its Essential Function in Fruit Breeding

The innovative technique known as Quantitative Trait Locus (QTL) mapping is crucial to the complex field of fruit breeding. According to Eduardo et al. [12], this genetic method painstakingly pinpoints specific areas of the genome that are closely associated with differences in particular characteristics such as fruit shape, color, size, or flavor. The deep value is in understanding how the genes controlling these characteristics are coordinated, how they interact with the environment, and how they can be used to improve important characteristics like fruit quality, productivity, and disease resistance [13].

The ability of QTL mapping to identify numerous genes causing trait variation is by far its greatest benefit. García-Gómez et al. [14] emphasized the abundance of genetic information available for interpreting the intricate connections between these genes and environmental factors.

These discoveries open the door to the development of accurate and successful breeding techniques, most notably MAS. According to Eduardo et al. [15], this tactic entails choosing plants that have particular genetic markers linked to desired traits, which improves and streamlines the breeding process.

Moreover, QTL mapping is also capable of discovering genes associated with numerous phenotypes, which is an important phenomenon when breeding plants with a variety of desired characteristics [16]. According to Carrasco-Valenzuela et al. [17], the combination of QTL mapping and MAS demonstrates how genetic insights and real-world breeding applications can work together to select plants that are likely to produce offspring that exhibit desired combinations of traits, which represents a paradigm shift in precision breeding.

In fruit crops other than peaches, QTL mapping has proven to be very helpful in identifying the genetic determinants of commercially significant features. For instance, QTL mapping has identified loci in grapes associated with berry size composition and resistance to powdery mildew—all of which are critical for the production of wine and table grapes [18]. Comparably, it has helped identify loci in apples linked to fruit firmness, shelf life, and resistance to diseases like apple scab [19]. QTL mapping has been crucial in the citrus industry for locating loci associated with fruit acidity, sweetness, and resistance to citrus greening disease, a severe worldwide issue [20,21]. Additionally, QTL mapping in peaches has improved our understanding of traits other than fruit quality like resistance to biotic stressors like fungi and aphids and abiotic stressors like dryness and freezing temperatures. These results emphasize how versatile QTL mapping is as a method for identifying and exploiting genetic variation in a variety of fruit crops and characteristics [22].

#### Handling Genetic Variation and the Difficulties of QTL Mapping: Fruit Color

The fascinating realm of fruit color, a characteristic that is closely linked to high genetic variation, presents difficulties for QTL mapping efforts [23]. The constant interplay between environmental and genetic variables determines the fruit color pallet [24]. Identifying the genetic areas that influence particular traits, such as fruit color, using QTL mapping requires careful analysis of genetic variation. Incorrect connections between genetic loci and the desired attribute may result from a failure to take this variation into consideration [25].

Anthocyanin pigments produce interesting hues that shed light on the genetic basis of fruit color. According to Chagné D et al. [26], the field of fruit color genetics is complicated since the generation of anthocyanins is regulated by a complex gene network. A range of fruit colors can be produced by genetic differences in these genes, which can lead to varying degrees of anthocyanin synthesis. The MYB10 gene mutation, which is associated with a yellow-green fruit color in apples, is particularly noteworthy as it provides evidence of the complex genetic interactions that shapes fruit aesthetics [11].

Researchers use sophisticated techniques like high-density genetic markers and Genome-Wide Association Studies (GWAS) to traverse the complexity of genetic variation. By taking into consideration population structure and kinship, the methods described by Cao et al. [27] and Zhang et al. [28] ensure a detailed understanding of the genetic loci responsible for fruit color and other crucial fruit traits.

### 1.2. Overview and Economic Significance of Peach Fruit Shape

Peaches come in a variety of morphologies from doughnut-like outlines to flat saucer shapes and represent a contrast between freestone and clingstone types [29]. Not only are peaches visually appealing but they are also a great source of vitamins minerals and antioxidants. Regular consumption promotes heart and digestive health and reduces the risk of some diseases [30].

A distinguishing and commercially significant characteristic of peach fruit is its tapestry-like shape, which is affected by a complex interplay of genetic and environmental influences. Fruit shape has a big impact on practical elements including harvest susceptibility, shipment endurance, and processing efficiency, even beyond how customers perceive it aesthetically [2]. Breeders and growers are focusing more on improving peach fruit shape due to the growing demand for visually appealing and high-quality peaches in the global market [12,31].

Recent advances in genetics have shown the genetic basis of peach fruit shape, mainly through QTL mapping [4]. According to Elsadr H. [32], QTLs linked to roundness, sphericity, and elongation are very useful for choosing plants that have the necessary fruit-shape traits and for breeding new cultivars that have better aesthetic appeal.

### 1.3. The Goal and Scope of the Review Is a Thorough Investigation of QTL Mapping for Peach Fruit Shape

In order to provide a comprehensive assessment of the current state of knowledge and possible future trajectories in QTL mapping for peach fruit shape, this paper sets out on a comprehensive trip. The complex interplay between genetics and environmental factors that define peach fruit shape is highlighted, elucidating the critical significance that fruit shape plays in the peach business. This review looks at the advancements in finding QTLs associated with peach fruit shape using data from a range of reliable scientific sources.

Simultaneously, this review broadens its scope to include an in-depth investigation of the nutritional content and health advantages that peaches possess. With the goal of serving as an invaluable tool for scholars, cultivators, and breeders, the review makes use of an in-depth comprehension of the genetic and environmental aspects that impact fruit shape. QTL mapping is a powerful technique that directs peach breeding efforts toward higher quality and production.

## 2. Quantitative Trait Loci (QTL) Mapping Journey: Exploration of Peach Fruit Shape

### 2.1. A Historical View on the Development of Peach Breeding Through QTL Mapping

A critical stage in the development of peach breeding techniques has been reached with the identification of the QTLs controlling peach fruit shape. Phenotypic observations were the foundation of traditional breeding techniques in the past; however, they are prone to subjectivity and environmental factors. According to Rawandoozi, Z.J et al. [33], the development of QTL mapping allowed for the identification of genetic areas linked to particular traits, which brought objectivity.

This change in strategy led to a more sophisticated understanding of the genetic architecture underlying the morphology of peach fruits, as noted by Veerappan et al. [34]. The robustness of this approach in elucidating basic genetic factors was highlighted by Hernández Mora et al. [31], who reported the consistency of several QTLs across various populations.

The process of QTL mapping for peach fruit morphology presents several challenges. Font i Forcada et al. [35] highlighted that in order to guarantee the production of reliable data, a careful selection approach for population mapping is required due to the stability and resolution of QTL effects. Environmental factors also make data accuracy more difficult, necessitating a comprehensive analysis of the interaction between genetic diversity, the environment, and the trait’s heritability [31].

### 2.2. Overcoming the Difficulties: QTL Stability and Settlement

#### 2.2.1. QTL Stability

The environmental conditions and genetic composition of the mapping population have a significant impact on the stability of QTL effects. The inconsistency of QTLs in various genetic settings and environmental circumstances can complicate the interpretation of results [33,36].

##### Stability of QTL Effects Across Environments

Effective breeding programs require the identification of stable QTLs across environments. Because they retained their shape under a range of environmental conditions, QTLs associated with fruit size, shape, and firmness were used in marker-assisted selection (MAS) as illustrated in Figure 2 [37].

This figure shows the distribution of QTL stability scores for traits like fruit size, shape, and firmness under various conditions. Within each box, the median is represented by the central line while the box bounds display the interquartile range (IQR), which includes the middle 50% of the data. There appears to be variation between conditions as the whiskers’ lowest and highest values fall within 1.5 times the IQR. The general consistency of these characteristics is shown in this illustration.

#### 2.2.2. QTL Resolution

QTL mapping accuracy is limited by the size of the mapping population and the density of molecular markers. Multiple genes may be included in QTL regions, making it difficult to pinpoint a single gene that causes a particular trait [38].

A difficult challenge in the QTL mapping process for peach fruit shape is choosing a suitable population that can produce reliable data for mapping [4]. The environmental conditions in which the population is cultivated further complicate the accuracy of the statistics. Given the complex interactions between the environment, genetic diversity, and the heritability of the trait, a careful selection procedure is essential [39].

### 2.3. Difficulties in Interpreting QTL Mapping Results for Peach Fruit Shape

Due to the complex genetic foundation of peach fruit shape, there are challenges in interpreting QTL mapping data. It is difficult to distinguish and pinpoint the precise effects of each QTL due to the intricate relationships between several genes and environmental factors [40,41].

The caliber and number of markers used are critical factors that influence QTL mapping’s correctness and dependability. Variations in marker usage may give rise to discrepancies in findings among research, making it difficult to compare and validate outcomes among various study groups [15].

In spite of these difficulties, QTL mapping has greatly improved our knowledge of the genetic basis of peach fruit morphology. Ongoing developments in genotyping technology, including whole-genome high-density SNP arrays and sequencing, hold the potential to improve QTL mapping accuracy and precision for this complex trait. Though the trip may be difficult, the knowledge acquired is priceless [42].

## 3. Techniques Exposed: Recent Peach Fruit Shape QTL Studies

QTL mapping is an important tool for understanding the subtle genetic traits associated with shape, drawing on the fundamental concepts of peach breeding.

### 3.1. Exposing the Methods: Current Research on QTLs for Peach Fruit Shape

Several approaches have been used in recent studies of peach fruit shape QTLs, each of which has illuminated a different aspect of this intricate characteristic. Using high-density SNP arrays and sequencing, Bielenberg et al. [43] were able to identify a number of important QTLs. Interestingly, a reliable association between chromosome 6 characteristics and fruit shape was found to be true in a variety of settings and years.

A different strategy was used to find numerous QTLs related to fruit morphology using a multi-parent advanced generation inter-cross (MAGIC) population [44]. One remarkable discovery was a QTL on chromosome 7 that was closely linked to the length and width of the fruit. Bulked segregant analysis and RNA sequencing were used by Zeballos et al. [45], who found probable genes and QTLs associated with the morphology of peach fruits, including one gene implicated in cell wall development.

The complex interactions between genetic and environmental elements that shape peach fruit continue to be mysterious despite these advancements [46].

Researchers found many important QTLs and candidate genes linked to the shape and form of fruit cell walls using a variety of methods including bulked segregant analysis, high-density SNP arrays, MAGIC populations, and RNA sequencing [47,48]. The complex interactions between genetic and environmental factors are still unclear despite these developments, underscoring the need for more study [46]. Table 1 lists the primary QTLs for peach fruit size, shape, and hardness, in addition to information on their chromosomal locations and environmental stability.

### 3.2. Navigating Complexity: Consistent and Generalizable Across Studies and Regions

The evaluation of our understanding of peach fruit shape QTLs reveals the intricate dynamics of generalizability and consistency. Aranzana et al. [51] investigated the consistency of fruit shape QTLs in a range of peach populations and discovered that certain QTLs were consistently associated with fruit shape traits across a variety of populations.

However, Martínez-Martínez et al. [52] revealed the difficulties in generalizing fruit shape QTLs across diverse Chinese areas. Although some QTLs were consistently associated with specific fruit shape traits, the study discovered that the strength of these associations differed by region. The intricate web of genetic and environmental interactions is further complicated by this, which presents fascinating questions about how environmental conditions impact the genetic basis of fruit shape.

Although some fruit-shape QTLs are consistent across studies and regions, the intricate interaction of genetic and environmental factors complicates matters. More research is needed to understand these characteristics and how they affect the consistency and generalizability of QTLs across populations and habitats.

### 3.3. Reducing the Gaps: Resolving Inconsistencies in the Knowledge Base

Notably, QTLs associated with peach fruit morphology have been identified, yet the present knowledge base still has significant gaps and inconsistencies. Significant information is lacking regarding the functional mechanisms underlying the identified QTLs. Genetic enhancement and selective breeding efforts are hampered by the lack of knowledge regarding the precise genes and mechanisms underlying the QTLs associated with fruit shape characteristics.

An additional layer of complexity is introduced by the inconsistent nature of QTL mapping findings. Certain QTLs are exclusive to certain populations or situations, whilst others exhibit consistency across studies and regions. This makes it more difficult to incorporate QTL mapping data into breeding strategies that work with a variety of populations and environments.

#### Methods for Covering Knowledge Gaps

**Comprehending Functional Processes:** Deciphering the functional mechanisms underlying QTLs continues to be an extremely difficult task. It is essential to apply functional genomics techniques like transcriptomics, proteomics, and metabolomics in order to identify potential genes and pathways that are involved in the regulation of fruit shape [53].

**Genetic Background Variability:** Results from QTL mapping may differ due to differences in the genetic backgrounds of the mapping populations. It is possible to increase the likelihood of finding consistent QTLs across diverse populations and environments by using larger mapping populations that show more genetic diversity [45].

**Standardizing Methods for QTL Mapping:** The range of phenotypic traits and QTL mapping techniques present difficulties. Enhancing study comparability will be possible through the use of uniform phenotypic trait assessment techniques [54].

**Combining Various Data Sources:** Combining data from several sources, such as transcriptome, phenotypic, and genomic data, is necessary to close gaps and resolve discrepancies. When synthesizing large-scale data, machine learning techniques can be extremely helpful in identifying new QTLs and potential genes that are involved in the regulation of fruit shape [55,56].

The consistency and generalizability of the traits ultimately depend on a comprehensive understanding of the underlying genetic and environmental factors, but a wide range of methodologies is also required to completely comprehend the complexities of peach fruit shape QTLs. As the gaps found highlight, further research, the application of cutting-edge techniques, and cooperation will be necessary to uncover the complex genetic fabric that shapes peach fruit.

## 4. Peach Fruit Shape: Exploring Unknown Ground in Future Orchards

### 4.1. Setting New Directions: Creative Approaches to Accuracy QTL Charting

Exploring the genetic subtleties of peach fruit shape requires not only an understanding of QTLs but also overcoming existing challenges. Promising directions and innovative approaches abound for future research that could fundamentally alter the accuracy and resolution of QTL mapping Figure 3 [57]. This circular workflow demonstrates how genomic selection, marker-assisted selection (MAS), and CRISPR are combined to improve peach breeding efficiency and accuracy.

#### 4.1.1. Novel Approaches to Accurate QTL Mapping

In order to identify more comprehensive genetic variation underlying peach fruit shape, advanced sequencing technologies such as high-volume sequencing, Oxford Nanopore PacBio, and Illumina can be employed [58]. Due in part to high-throughput sequencing’s depth and speed, QTL mapping has better resolution [59].

**Bioinformatics Tools:** Gene ontology databases, genome browsers, and gene expression analysis software are just a few of the tools that researchers can use to sort through mountains of genetic data. By finding potential gene candidates linked to QTLs, these methods are essential in advancing our knowledge of fruit shape regulation [60,61].

**Phenotyping Techniques:** Modern phenotyping methods like computer vision and 3D imaging offer a rapid and precise method of measuring different fruit shape attribute claims [62]. These techniques can expedite QTL mapping and offer a more thorough understanding of the genetic landscape.

**Multi-parental Mapping Populations:** By deliberately combining several parental lines, a heterogeneous population is produced, which improves QTL mapping’s resolution and variety. QTLs with broader and more consistent effects are found using techniques like nested association mapping and multi-parent advanced generation intercross populations [63].

**Combining genome-wide association studies (GWAS) with QTL mapping:** GWAS and QTL mapping work together to uncover genetic variations amongst a variety of populations. This combination broadens the investigation to genetic variants outside of the QTL region while also improving our understanding of consistent QTL effects [64].

**Gene editing using CRISPR/Cas9:** Using CRISPR/Cas9 gene editing tools enables accurate genome manipulation. CRISPR/Cas9 is a potent technique for validating candidate genes and revealing their functional significance in fruit shape regulation, even though it raises ethical questions [65].

For CRISPR-edited crops, there are discernible local differences in the regulatory environment. In the US, regulations are more relaxed, which promotes quick innovation. Stricter regulations are enforced by the European Union, which designates crops modified by CRISPR as genetically modified organisms (GMOs). Furthermore, since public opinion greatly affects legislation, it is crucial to communicate openly about the advantages and security of CRISPR-based technology [66].

Traditional methods are changing as a result of the use of genomic techniques like GWAS and CRISPR/Cas9 in peach breeding. For instance, direct control of the genes governing important characteristics like fruit size and quality has been made possible by CRISPR-based precision breeding. This strategy guarantees quicker trait selection while preserving genetic diversity when used in conjunction with marker-assisted selection (MAS). Additionally, breeders can assess several quantitative traits at once, like disease resistance and climate change adaptability, thanks to genomic selection (GS), which speeds up the breeding process as a whole [67,68].

**Tool Integration:** By combining CRISPR/Cas9 with GWAS, MAS, and GS, a comprehensive understanding of contemporary peach breeding methods is produced [69].

**Viewpoint on Regulation:** The subject gains depth and importance when disparities in public opinion and regulation (such as those between the US and the EU) are brought up, demonstrating an understanding of global viewpoints [70].

**Quantitative attributes:** The case for GS’s acceptance is strengthened by highlighting its capacity to address multiple traits at once [71].

#### 4.1.2. Future Breeding: Marker-Assisted Selection, Genomic Selection, and Beyond

Using marker-assisted selection (MAS) and genomic selection (GS) seems like a viable way to influence peach breeding in the future.

Marker-Assisted Selection (MAS), Genome Selection (GS), Genetic Engineering, and CRISPR/Cas9:

Using molecular markers connected to particular QTLs, MAS streamlines the selection procedure while cutting down on the expense and time required for conventional phenotypic approaches (Figure 4). MAS has the potential to greatly increase breeding operations’ accuracy and effectiveness [72].

Using high-throughput sequencing and advanced statistical models to assess each individual’s genetic composition, GS expedites the breeding process and improves the effectiveness of peach breeding programs. The identification of peaches with the desired fruit shape phenotypes can be sped up by integrating GS [73,74].

Research on transgenic methods and genome editing tools according to Windham, J.K. [75], opens up new possibilities for altering the genes that determine fruit shape. While precise genome modification is possible with CRISPR/Cas9, transgenic techniques employ foreign genetic material. Nonetheless, both strategies require thorough ethical analysis and legal frameworks [76].

Figure 4A shows the common breeding techniques of the future that employ marker-assisted selection and CRISPR/Cas9 genomic selection. In order to emphasize the revolutionary potential of genomic-assisted breeding techniques, significant metrics are compared in Figure 4B. Although genomic-assisted methods make use of developments in high-throughput sequencing, predictive modeling, and marker-based trait selection, traditional phenotypic selection data represent the accuracy and typical durations found in long-term field trials [75,76]. Faster selection processes are made possible by MAS and GS, which significantly reduce the breeding cycle time in genomic approaches. Breeding efforts that give priority to fruit size yield and abiotic stress resistance are the basis for the trait improvement scores. These comparisons show how genomic-assisted approaches are more precise, scalable, and effective in meeting market-driven demands and climate change, while also highlighting crucial elements like breeding cycle length, accuracy, and environmental resilience.

### 4.2. Overcoming Obstacles: Limitations in the QTL Mapping Journey

Notwithstanding QTL mapping’s promises, obstacles still exist that must be carefully avoided.

#### The QTL Mapping Expedition’s Restrictions

Peach Fruit Shape Complexity and Absence of a Robust Reference Genome:

The intricate interplay between genetic and environmental factors that determine fruit shape is a major issue. Finding and characterizing QTLs remains a challenging task, despite this complexity [77,78]. Due to the absence of a reliable peach reference genome, QTL mapping accuracy is limited. The development of such a resource is ongoing with the aim of increasing QTL analysis and discovery accuracy [79,80].

### 4.3. Moving Forward: Outlining the Future Research Environment

#### Outlining the Prospects for Future Research

Extend QTL Discovery, Recognize Molecular Mechanisms, Accept Innovation in Phenotyping, Investigate how genes and environments interact, and Optimize Breeding Techniques:

Research should concentrate on further examining and validating QTLs associated with peach fruit morphology. Using more sophisticated statistical methods, such as genome-wide association studies and mixed-model approaches, and larger populations can uncover new facets of genetic influence [81].

It is critical to investigate the molecular mechanisms underpinning QTLs that have been found. By figuring out how these genes affect peach fruit morphology, breeding programs can be more focused and successful while also perhaps revealing new genes and pathways [82].

High-throughput phenotyping techniques, in conjunction with sophisticated imaging and machine learning methodologies, present promising opportunities for the accurate and timely evaluation of fruit shape attributes. Research can be streamlined by incorporating these improvements into QTL mapping projects [83,84].

Examining how genetic and environmental factors interact dynamically to shape peach fruit morphology can help develop more resilient and adaptable cultivars. It is easier to adapt fruit kinds to particular development conditions when one is aware of how external influences influence fruit morphology [1].

There are several opportunities to investigate the possible synergy between QTL mapping genomic selection and marker-assisted selection to improve peach flavor texture and disease resistance among other qualities. By referencing their effective use in other crops, it is possible to illustrate the potential of these breeding techniques for peaches [85,86].

### 4.4. Beyond Boundaries: QTL Mapping’s Potential Effects on Peach Cultivation

Despite the considerable progress made in QTL mapping for peach fruit morphology, more work is anticipated in the future. Utilizing new technologies, refining breeding techniques, and expanding our knowledge of the biological processes influencing fruit shape are all ways that researchers can increase peach production [18,33].

#### QTL Mapping’s Potential Effects on Peach Cultivation

To match customer expectations and maintain competitiveness in the market, fruit quality, productivity, and marketability improvements hold the key. Future peach orchards beckon, full of opportunities for anyone willing to explore the unexplored territory of QTL mapping [46]. The flowchart in Figure 5 provides a concise and comprehensive overview of the QTL mapping procedure for the morphology of peach fruit. It describes the process that starts with gathering genotyping and phenotypic data, including genomic selection (GS) and multi-environment trials, and ends with introducing improved cultivars. Fruit quality productivity and marketability are all increased by this systematic approach, which also increases the precision and efficacy of contemporary breeding programs.

Quantitative trait loci (QTLs) affecting important peach fruit shape factors have been painstakingly uncovered by several research studies. Certain fruit shape traits, for example, have been shown to be significantly impacted by QTLs on chromosomes 1, 3, and 4, with a particularly important QTL on chromosome 5 being closely associated with the fruit length and shape index. Through careful parent crossing, these discoveries equip breeders with the necessary knowledge to produce new cultivars that include enhanced fruit shape attributes such as symmetry, unishapeity, and appeal to consumers.

In addition to revealing genetic components influencing fruit shape, QTL mapping has made a substantial contribution to our comprehension of the complex genetic mechanisms at play. Prominent identifications of candidate genes within QTL intervals linked to hormone signaling and cell wall production underscore possible biological mechanisms governing the development of fruit shape.

Beyond new scientific understanding, improved fruit shape characteristics can have a big impact on the peach business. Growers and retailers who produce homogeneous, eye-catching peaches may be able to charge higher prices. Through QTL mapping and improved breeding techniques, cultivars with beneficial QTL alleles may show increased pest resistance and adaptability to certain conditions, hence boosting profitability and sustainability.

## 5. Conclusions

Research on peach fruit shape is essential for improving fruit quality and marketability. Breeders now have efficient tools for crop improvement due to the development of QTL mapping, which has equipped breeders with powerful tools for crop improvement, enabling a deeper understanding of the genetic foundations of complex traits. In order to advance marker-assisted selection (MAS) and genomic selection (GS) in peach breeding programs, this review emphasizes the discovery of stable QTLs for crucial traits such as fruit shape, size, and firmness. Furthermore, promising solutions are provided by cutting-edge methods such as transcriptomics and CRISPR/Cas9 gene editing that address complex trait regulation and genotype–environment interactions. These advancements significantly enhance the precision and efficiency of breeding efforts since they address critical knowledge gaps in peach fruit morphology.

Machine learning for multi-omics data analysis and genome-wide association studies facilitates faster breeding cycles and the development of resilient peach cultivars that can adjust to changing market demands and climatic challenges. In conclusion, a revolutionary development in the study of QTL mapping for peach fruit shape is imminent. By overcoming the current challenges and producing improved peach cultivars, researchers can ensure the long-term viability and competitiveness of the global peach market with the aid of contemporary genomic technology and interdisciplinary collaboration.

## Figures and Tables

**Figure 1 plants-14-00175-f001:**
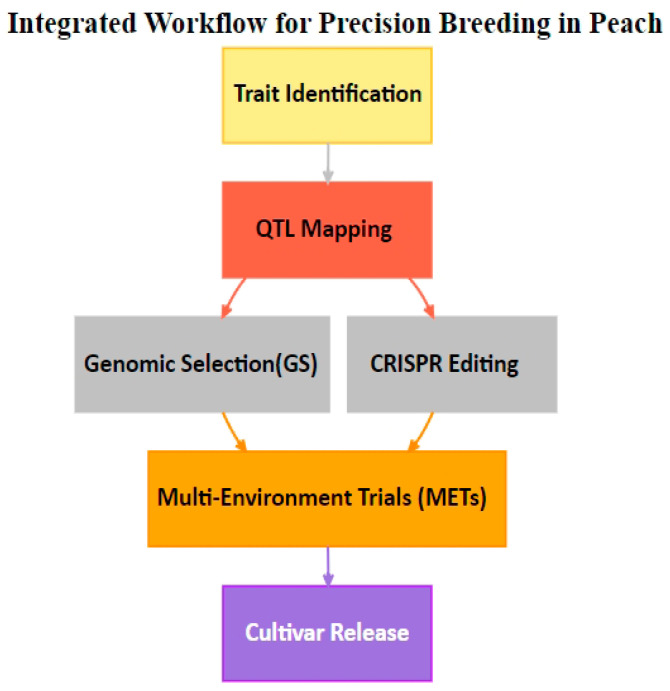
Enhancement through CRISPR and QTL mapping.

**Figure 2 plants-14-00175-f002:**
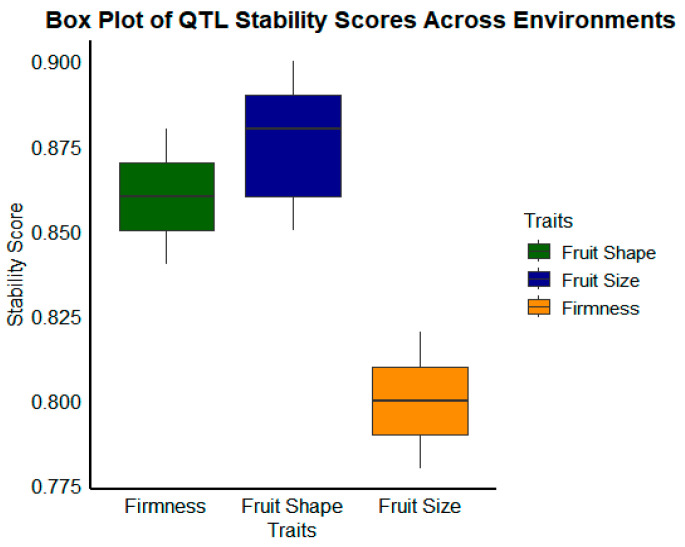
Shape of important QTLs in different environments.

**Figure 3 plants-14-00175-f003:**
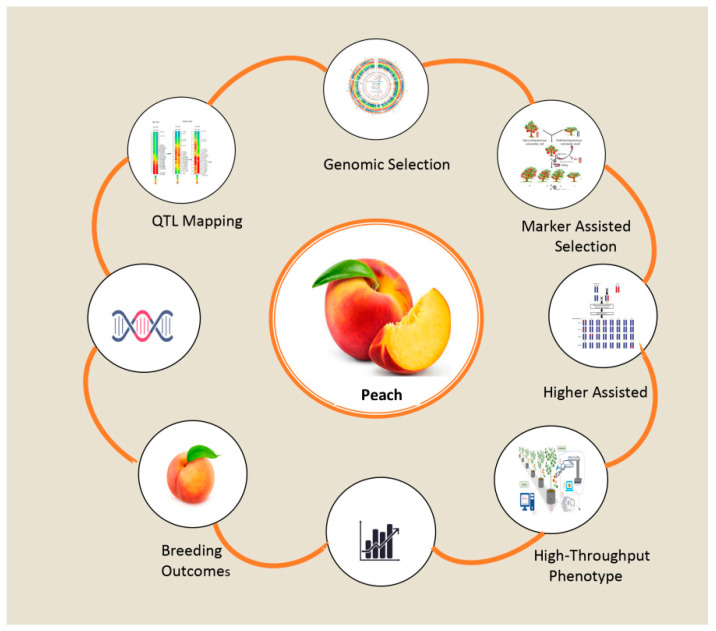
Process for using advanced methods in peach breeding.

**Figure 4 plants-14-00175-f004:**
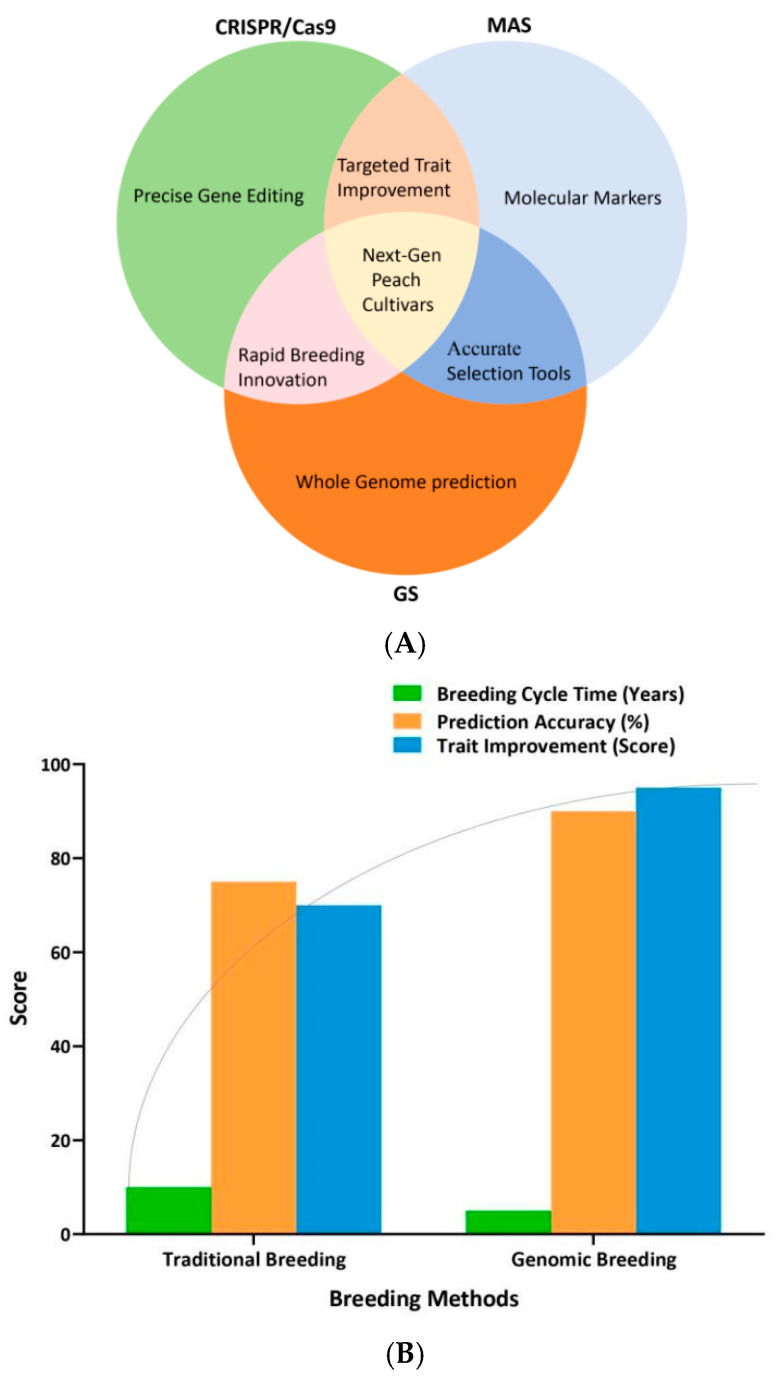
Examining and contrasting conventional and genomically assisted breeding methods. (**A**). Depicts common future breeding approaches that use CRIPSR/cas9 and genomic selection marker-assisted selection. (**B**). Comparing and assessing traditional and genomically assisted breeding procedures.

**Figure 5 plants-14-00175-f005:**
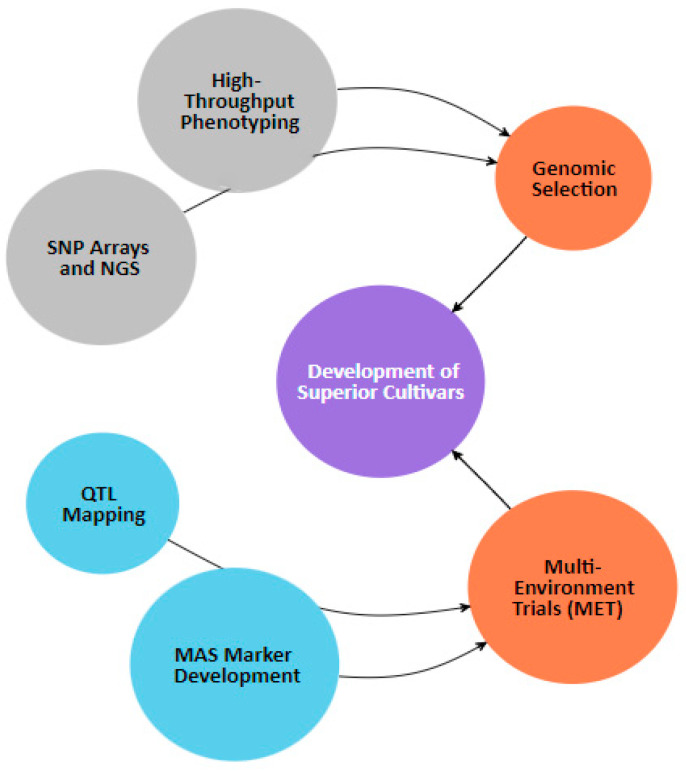
This flowchart provides a concise and comprehensive overview of the QTL mapping procedure for the morphology of peach fruit.

**Table 1 plants-14-00175-t001:** Important QTLs for peach fruit shape and related traits.

Name and Region of QTL	Related Traits of Fruit	Location on Chromosomes	Consistency	References
QTL-1.5	Size	1	Very stable in a variety of contexts	[15]
QTL-2.4	Firmness	2	Higher	[49]
QTL-3.1	Spherical shape	3	Stable to a moderate extent	[27]
QTL-5.3	Roundness	5	Consistent in three environments	[50]
QTL-7.2	Elongation	7	Consistent in two different environments	[45]

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
