# Peer review of "Using Quantitative Trait Locus Mapping and Genomic Resources to Improve Breeding Precision in Peaches: Current Insights and Future Prospects"

_plants, 2025, doi:10.3390/plants14020175_

Round 1
Reviewer 1 Report
Comments and Suggestions for Authors
The review entitled “Using Quantitative Trait Locus Mapping and Genomic Resources to Improve Breeding Precision in Peaches: Current Insights and Future Prospects” discuss the employment of QTL mapping and the innovative breeding techniques for the development of new peach cultivars with a particular focus on fruit shape.
In general, the paper is well written and clear, although there are some redundancies, and both the main text and figures can be improved. Following my suggestions for the general improvement of the manuscript:
Title (lines 2-4): Being shape fruit trait a central argument in the manuscript, I would suggest including it in the title, as for example “the case of fruit shape” or “a focus on fruit shape”.
Line 59: Please, indicate “Prunus persica (L.) Batsch” with italic when needed instead of just “Prunus persica”.
Line 69: Is the word “people” a typo?
Figure 1: In order to be more readable, this figure should be rotated by 90°. Moreover, Figure 1 seems to be redundant with Figure 5. Please make it clearer, even making a single figure from Figure 1 and Figure 5.
Lines 83-103: Please, add more comments on the usefulness of QTL mapping for identifying the genetic determinism of trait of interest even in other fruit crops of economic importance (grape, apple, citrus) and in peach for traits different than fruit quality (e.g. resistance to biotic and abiotic stress).
Line 95: One marker-assisted selection acronym (MAS) was explained, please use just MAS in the text. The same for genomic selection (GS) and for all the other acronyms.
Line 124: This paragraph could be implemented with a more detailed discussion (a few sentences) on peach varieties commercially available (peach varietal panorama) and on consumer preferences, if available, aimed at the definition of a peach ideotype to be considered in peach breeding programs.
Lines 185-188 and Figure 2: Which data were used for making these boxplots? Caption is not clear and has to be improved. Moreover, why are red spots (line 187) mentioned if not present?
Table 1: Column “Fruits” is unnecessary.
Line 291: Please avoid the two dots in the paragraph title, especially if a sub-paragraph title is present then.
Line 361: Is the word “people” a typo?
Lines 368-370: Please, elucidate in the main text which data were used to realize the bar plot (Figure 4B) and make a comment on them.
Lines 412-413: Probably there is a typo in the references between square parentheses. Moreover, the paragraph title should be at the head of the line.
Figure 5: The text in the figure is too little and lines need arrows to make it more readable. Moreover, Figure 5 seems to be redundant with Figure 1. Please make it clearer, even making a single figure from Figure 5 and Figure 1.
Author Response
I genuinely appreciate your critical and in-depth assessment of my work. Your professional advice helpful suggestions and perceptive remarks have greatly raised the caliber and readability of this work. For your concerns to be successfully addressed each suggestion has been thoroughly thought out and put into practice. your detailed and comprehensive advises and suggestions enhanced the work and deepened my comprehension of the topic. I truly appreciate the time and effort you invested in reviewing my work and offering such insightful feedback. I’m hoping the revised version meets your exacting requirements and appropriately takes into account all of your insightful comments. I want to thank you once more for your important contributions to this project.

Reviewer 2 Report
Comments and Suggestions for Authors
The topic is relevant, there is currently a lack of articles linking practical selection and modern achievements in biotechnology, genomics and informatics. But the article has a number of shortcomings: the manuscript, at least the abstract and introduction, are not prepared according to the journal requirements. There are a lot of general statements and few specific examples and their discussion. The figures and table are not very informative and there is a lot of redundant information. More detailed comments are given below:
Title:
Using Quantitative Trait Locus Mapping and Genomic Resources to Improve Breeding Precision in Peaches: Current Insights and Future Prospects
If the article deals only with fruit morphology, this should be reflected in the title of the article
Abstract:
According to Plants Instructions for authors: The abstract should be a total of about 200 words maximum. The abstract should be a single paragraph and should follow the style of structured abstracts, but without headings. The submitted manuscript is sufficiently longer – over 340 words and structured not as single paragraph – needs to be corrected according to the journal's requirements. No references usually in the abstract text.
Introduction is to long.
Line 66 sentence “According to [11], genomic selection simplifies the process…” should be modified: “According D’Agostino and Tripodi [11] genomic selection simplifies the process…”
And same in other places of the manuscript text, except when the reference number is at the end of a sentence, or when there are separate citations in one sentence
Line 69 “peach” or “peaches” instead of “people”?
Line 69- 78 It should be noted that references [12] and [13] are devoted on rice and apple trees investigations, respectively. The location of reference [13] should be clarified.
Line 80 The Figure 1 is originally made by authors, or it from any reference?
Line 105 It would be more appropriate to say “high genetic variation”?
Line 112-113 “Fruits' captivating colors, which are caused by the pigment’s anthocyanins, reveal the genetic basis of fruit color”. I don't understand what the authors wanted to say.
Line 121 instead of “supported” might be “described” or “ applied”
Line 147-149 The sentence “Investigating the progress made in identifying QTLs linked to peach fruit shape, the review consults a wide range of reliable sources, such as peer-reviewed journals and respectable publications” seems redundant.
Line 187 Red spots is invisible in figure 2
Line 238 Table 1 should be modified: Second column could be named “related traits of fruit” then you shouldn't repeat the word fruit in the column. Column “location on chromosomes” it would be enough to write the chromosome number, without the word “chromosome”. Moreover, the chromosome number is clear from the first column. The sixth column is completely unnecessary because it is uninformative.
Line 240-241 Sentence “When assessing our understanding of peach fruit shape QTLs, the subtle dynamics of consistency and generalizability become clear” is difficult to understand and should be modified.
Line 297 should be comma after words “genomic selection”.
Line 376 Figure 4B is unclear – what means black line? How calculated “trait improvement” score?
Line 413 comma between reference number 80 and 81.
Line 415-417 Sentences like this: “Researchers can improve peach production by utilizing new technologies, improving breeding methods, and learning more about the biological processes that affect fruit shape [30, 82].” Is like from textbook and self-explanatory.
Line 428 Figure 5 The sides of the picture should be swapped - the target “development of superior cultivars” should be on the right side.
Line 431-437 should be citations in this paragraph
Comments on the Quality of English LanguageAuthors should review the text and correct errors
Author Response
I want to start by sincerely thanking you for reviewing my manuscript. I value your advice and insightful remarks which have improved my writing abilities and academic writing proficiency.
“I want to sincerely thank Reviewer 1 for their insightful remarks and the time they invested in giving me thorough feedback on my manuscript. Their perceptive comments and observations were not only helpful and legitimate but they were also crucial in raising the caliber of my manuscript and honing my academic writing abilities. I made sure that every point of the suggestion was taken into account by carefully addressing each comment and implementing the recommended changes throughout the manuscript. It is my earnest hope that the changes will satisfy the reviewers requirements and sufficiently address their issues. Once more I sincerely appreciate their efforts to improve the works rigor and clarity and I am appreciative of their contributions to the research improvement”.
Thank you so much.
Sincerely,
UMAR HAYAT

Reviewer 3 Report
Comments and Suggestions for Authors n this study, investigation of the genetic basis controlling peach fruit shape is undertaken in is explored. QTL mapping, marker-assisted selection, genomic selection, and novel genetic interventions are detailed. The paper seeks answers to important questions, the goal is to facilitate the understanding the possibilities of breeders. Please at least at the first appear, mention the scientific name of peach. Graphs are sufficiently detailed and easy to expound. Reference of Figures are missing. The text contains occasional typos, but it is nevertheless easy to read. The English of the manuscript is acceptable, but the academic accuracy can be improved, wording is poor. The literature used is not much, but enough. The whole manuscript seems superficial.Author Response
My sincere appreciation goes out to Reviewer 3 for their comprehensive and perceptive feedback on our manuscript. This work's scholarly excellence, clarity, and organization have been greatly improved by your insightful insights and practical suggestions. As it has helped us refine significant portions of the work, we really appreciate you’re your cooperation. We have deepened our analysis and gained valuable knowledge by responding to your comments, which will inform our future research endeavors. Once more, I would want to express my gratitude for your valuable time and significant contributions to this work. We sincerely hope the new version meets your needs.
We believe these modifications enhance the clarity, rigor, and completeness of our manuscript. We are confident that our responses address the reviewers' concerns and improve the overall quality of the manuscript.
Thank you for considering our rebuttal.
*Missing a comment doesn't imply I disregarded it; rather, it indicates that it was one of the same reviews that I had already responded to.
Sincerely,
UMAR HAYAT

Reviewer 4 Report
Comments and Suggestions for Authors
Dear Authors,
The submitted review article is an interesting review exhibiting the progress in genetic mapping and QTL analysis in peaches, aimed at breeding purposes for cultivar improvement. The authors tried to explain the worthy review literature in detail form but still need major improvements.
*Up to the present time, a lot of research on molecular genetics and breeding basis studies for identification of functional and candidate genes has been published in peach crop that should be incorporated in the details.
*The writing language of the manuscript is not so good. It needs to be modified in extensive form.
*Figures quality is too low. It is strongly recommended to maintain the font size into visible format and upload the high-quality figures (300 dpi).
*The font size of all figures should be enlarged enough, labeled, and titles of figures and tables must contain the explanation of captions.
*Title: It must be modified in proper scientific style.
*Introduction: It is written too short and strongly recommended to add more and comparative studies focused on explaining the complex genetic regulatory mechanisms of crop-specific traits.
*Figures: It must need to be revised by drawing in your own way. The figure sections do not need to be copied and pasted. Insert your own approach.
Conclusion: It should be written in concise form by explaining the novel results of the experiment.
Comments on the Quality of English LanguageThe English could be improved (extensive editing) to more clearly express the research.
Author Response
I wish to thank you for your informative and thorough critique on my work. Their intelligent remarks, helpful recommendations, and guidance have greatly improved the quality of this work. After carefully considering each of their ideas, I implemented the necessary changes to ease their concerns. The comprehensiveness of their evaluation not only improved the manuscript, but also allowed me to understand more about the subject. I appreciate them taking the time to read my work, and I hope the amended manuscript meets their standards and incorporates all of their helpful feedback.
We believe these modifications enhance the clarity, rigor, and completeness of our manuscript. We are confident that our responses address the reviewers' concerns and improve the overall quality of the manuscript.
Thank you for considering our rebuttal.
*Missing a comment doesn't imply I disregarded it; rather, it indicates that it was one of the same reviews that I had already responded to.
Sincerely,
UMAR HAYAT

Round 2
Reviewer 2 Report
Comments and Suggestions for Authors
The authors have tried to improve the manuscript of the article, and they have partially succeeded. However, it still lacks systematicity and deeper analysis. The authors have not corrected some things, for example, the unnecessary sixth column in Table 1. The publisher must decide whether to publish this article.
Author Response
“I want to sincerely thank Reviewer 2 for their insightful remarks and the time they invested in giving me thorough feedback on my manuscript. Their perceptive comments and observations were not only helpful and legitimate but they were also crucial in raising the caliber of my manuscript and honing my academic writing abilities. I made sure that every point of the suggestion was taken into account by carefully addressing each comment and implementing the recommended changes throughout the manuscript. It is my earnest hope that the changes will satisfy the reviewers requirements and sufficiently address their issues. Once more I sincerely appreciate their efforts to improve the works rigor and clarity and I am appreciative of their contributions to the research improvement”.
Thank you so much.

Reviewer 3 Report
Comments and Suggestions for Authors
I think, that quality of the revised version improved a lot, therefore I suggest to accept it.
Author Response
Thank you, Academic Reviewer 3. I truly thank you for allowing me to publish my manuscript. Your astute critique and practical ideas considerably improved the quality of this work, and your skills helped me navigate the editing process. I sincerely appreciate you taking the time and effort to read my manuscript. Your insightful evaluation improved the paper and helped me grow as a researcher. I'd want to thank you again for your help and the opportunity to publish my work. Warm regards. Hayat Umar, Fruit Tree University of Zhengzhou. Agricultural Sciences Academy in China.

Reviewer 4 Report
Comments and Suggestions for Authors
The authors did not revise the review article in a serious manner.
To date, many QTL studies have been conducted that identified versatile QTLs using traditional and advanced genome-assisted approaches.
They just highlighted the random text in the manuscript and did not incorporate the suggested literature reviewer as well as other comments in detail as per the suggestion of the reviewer.
So, the manuscript is just wasting the time of the reviewer(s), and it is not suitable for further peer review. Hence, it is strongly recommended for rejection.
Thanks
Author Response
I'd like to express my heartfelt gratitude to the reviewer 4 who took the time and effort to assess our work, "Using Quantitative Trait Locus Mapping and Genomic Resources to Improve Breeding Precision in Peaches: Current Insights and Future Prospects" Their insightful remarks and constructive suggestions have really aided in the improvement of our work.
